# Role of Alternatively Spliced Messenger RNA (mRNA) Isoforms of the Insulin-Like Growth Factor 1 (IGF1) in Selected Human Tumors

**DOI:** 10.3390/ijms21196995

**Published:** 2020-09-23

**Authors:** Aldona Kasprzak, Witold Szaflarski

**Affiliations:** Department of Histology and Embryology, University of Medical Sciences, Swiecicki Street 6, 60-781 Poznań, Poland; witold.szaflarski@gmail.com

**Keywords:** alternative splicing, IGF1, various isoforms (RNA, protein), human cancers

## Abstract

Insulin-like growth factor 1 (IGF1) is a key regulator of tissue growth and development that is also implicated in the initiation and progression of various cancers. The human IGF1 gene contains six exons and five long introns, the transcription of which is controlled by two promoters (P1 and P2). Alternate promoter usage, as well as alternative splicing (AS) of *IGF1,* results in the expression of six various variants (isoforms) of mRNA, i.e., IA, IB, IC, IIA, IIB, and IIC. A mature 70-kDa IGF1 protein is coded only by exons 3 and 4, while exons 5 and 6 are alternatively spliced code for the three C-terminal E peptides: Ea (exon 6), Eb (exon 5), and Ec (fragments of exons 5 and 6). The most abundant of those transcripts is IGF1Ea, followed by IGF1Eb and IGF1Ec (also known as mechano-growth factor, MGF). The presence of different IGF1 transcripts suggests tissue-specific auto- and/or paracrine action, as well as separate regulation of both of these gene promoters. In physiology, the role of different IGF1 mRNA isoforms and pro-peptides is best recognized in skeletal muscle tissue. Their functions include the development and regeneration of muscles, as well as maintenance of proper muscle mass. In turn, in nervous tissue, a neuroprotective function of short peptides, produced as a result of IGF1 expression and characterized by significant blood-brain barrier penetrance, has been described and could be a potential therapeutic target. When it comes to the regulation of carcinogenesis, the potential biological role of different var iants of IGF1 mRNAs and pro-peptides is also intensively studied. This review highlights the role of IGF1 isoform expression (mRNAs, proteins) in physiology and different types of human tumors (e.g., breast cancer, cervical cancer, colorectal cancer, osteosarcoma, prostate and thyroid cancers), as well as mechanisms of IGF1 spliced variants involvement in tumor biology.

## 1. Introduction

The mature insulin-like growth factor 1 (IGF1), serving as a systemic growth factor, is a highly conserved circulating 70 amino acid (aa) single-chain basic polypeptide, acting as a master regulator of cell proliferation, differentiation, and apoptosis [1]. Together with insulin and IGF2, it belongs to the insulin superfamily, synthesized as prepro-proteins consisting of 4 domains (pre, B, C, A). In nuclear magnetic resonance (NMR) solution structure, IGF1 is a more structurally dynamic molecule [2] and has more potent mitogenic and anti-apoptotic activities than insulin [3]. The liver is the major site of IGF1 production, with growth hormone (GH) being the main regulator of its secretion (a negative feed-back loop) [4,5,6,7]. The hepatic IGF1 mainly exerts endocrine activity, while IGF1 synthetized by other tissues acts in a para- and/or autocrine way [8,9,10,11]. Biological activity of the mature IGF1 and its isoforms in physiology is mostly dependent on the phosphorylation of type I IGF1 receptor (IGF1R) [12,13]. In turn, in cancer cells, IGF1R-, insulin receptor (INSR)- and/or hybrid IGF1R/INSR-independent activity has also been described [14,15,16,17,18].

The human IGF1 gene is composed of six exons, four of which are alternatively spliced (AS) and is transcribed in a wide variety of normal and tumor cells [19,20,21,22,23]. The main AS types of the *IGF1* are exon skipping, intron retention, and alternative 3′ and 5′ splice sites [24,25]. IGF1 mRNA isoforms possess different stabilities, binding partners, and activity, indicating an important role of these molecules [21]. The mechanisms responsible for generation of various IGF1 mRNAs from pre-mRNA IGF1 molecules seem to be conserved among mammalian species [26,27,28]. In turn, alternative IGF1 mRNA splices generate different precursor peptides (IGF1 protein isoforms), but do not alter the structure of mature IGF1 protein [9,23,26,29]. Mitogenic and anti-apoptotic activity of mature IGF1, as well as different precursor IGF1 peptides, qualify IGF1 to the group of growth factors implicated in the initiation and progression of various cancers [16,30,31,32,33,34,35].

The knowledge of the basic functional effects of IGF1 gene AS could contribute to the discovery of new molecular mechanisms of carcinogenesis and, consequently, to the use of that knowledge in anti-cancer therapeutic approaches.

This review highlights the role of IGF1 isoform expression (mRNAs, proteins) in different types of human tumors, as well as the mechanisms of IGF1 spliced variant involvement in tumor biology.

## 2. Alternative Splicing (AS)—A Phenomenon in Physiology and Pathophysiology

AS is a highly controlled process which increases the coding potential of the genome and is well documented at the transcriptional level through cDNA sequence evidence, as well as microarray and RNA-seq experiments. However, large-scale proteomic experiments identify only few alternative isoforms resulting from this process [36,37,38,39,40]. AS enables the formation of multiple mRNA isoforms (variants) from a single gene, that may be translated into a range of diverse protein products [23,36,37,38,41]. It allows for transcriptomic plasticity, controlling which RNA isoforms are expressed in different cell types or time point [42]. The most commonly observed AS mechanisms are cassette alternative exon (included or skipped in the transcript), alternative 5′ splice sites, alternative 3′ splice sites, intron retention, mutually exclusive alternative exons, alternative promoter and first exon, and alternative poly A site and terminal exon [19,21,23,37,43]. It is estimated that over 15,000 AS events are linked to different aspects of tumor biology, as well as sensitivity and resistance to chemotherapeutics [44].

Multifactorial AS regulation is often emphasized, both in normal and transformed cells [42,44]. In cancers, transcriptomic changes induced by AS, as well as mutations in splicing factors of specific cancer genes or in the regulatory splicing machinery, are also described [41,42,45,46,47]. Recently, hypoxia-induced AS was even considered the 11th hallmark of cancer and a very important determinant of therapeutic resistance [48]. Impaired pre-mRNA splicing can lead to altered gene expression patterns and result in cell transformation and carcinogenesis. Hence, cancer often results from pre-mRNA splicing abnormalities, with many isoforms preferentially re-expressed in tumors [45].

The signal transduction pathways that affect AS of gene transcripts being important in carcinogenesis include, among others, the IGF signaling pathway. While the role of mature mRNA IGF1 molecules, as well as precursor forms of IGF1 protein in physiology (development and aging) as well as pathology (muscular dystrophy, neurological disorders, and cancer), is of key importance, its molecular mechanisms are largely undiscovered [21,23,29,33,35,49].

## 3. IGF1 Gene Structure and Signaling

The structurally heterogeneous IGF1 gene (~85.1 kb) exists in a single copy in the human genome, located on the long arm of chromosome 12 (12q23.2) [50,51]. It has 6–9 transcripts (splice variants), 263 orthologues, three paralogues, is associated with three phenotypes and belongs to one Ensembl protein family [24,51]. The nucleotide sequence of human liver cDNA encoding the complete aa sequence of IGF1 was first reported by Jansen et al. [52]. Similar to rat *Igf1,* the human *IGF1* is composed of six exons and five introns [26,53]. It was recently reported that an additional exon is present in the humans (mainly in the liver), located upstream from exon 1. This fragment can be spliced directly into exon 3, or, less often, into exon 1 [54].

IGF1 can act through a number of specific glycoproteins membrane receptors, including IGF1 type I receptor (IGF1R), type II (IGF2R), insulin receptor (INSR) and hybrid receptors (IGF1R/INSR). Most of the effects of IGF1 in physiology and pathophysiology is mediated by IGF1R and canonical signaling pathways such as the Phosphatidylinositol 3-Kinase (PI3K)/Serine-threonine Protein Kinase (Akt) and Ras/Raf/Mitogen-activated Protein Kinase (MAP) pathways. Furthermore, the IGF1/IGF1R axis is also an important pathway involved in cell transformation [1,11,13].

### 3.1. IGF1 Gene Promoters

Based on animal models and in vitro cultured cells, it was proven that, in mammals, the transcription of *IGF1* remains under control of two promoters: P1 and P2, located before, respectively, exon 1 and exon 2 [55,56,57,58,59,60,61]. In most of the human and animal tissues, the majority of transcripts are generated from P1 (over 90% transcripts), with this promoter appearing to be far more active than P2 [59,62,63,64].

Both promoters are TATA-less, CCAAT-less, lacking defined transcriptional start points, as well as GC-rich areas or CpG-islands with a heterogeneous transcription initiation. As a result, the initial portion of the 5′ untranslated region (UTR) of human *IGF1* exon 1 is required for high level of P1 basal transcription [59,65]. The 5′UTR is highly conserved in human, as well as other mammalian genomes (mouse, rat, cattle, sheep), with the first 412 bp of the 5′ flanking region responsible for basal promoter activity of exon 1. Transcripts of different signal peptide structure are generated using four transcription sites of exon 1 [21,52].

Recent human and macaque studies confirm the notion that the biggest amount of IGF1 transcripts produced with participation of both promoters is located in the liver, as well as in both white and brown adipose tissue. Interestingly, adipose tissue is characterized by a 5–6-fold higher amount of IGF1 mRNA compared to liver [64]. Moreover, IGF1 promoter activity shows a cell type-specific expression pattern [66]. Rat model studies showed varying influence of GH on hepatic chromatin in the *IGF1* locus and, consequently, *IGF1* transcription in the liver by distinct promoter-specific mechanisms [67].

Considering alternate promoter usage (different leader sequences), two different classes of IGF1 mRNA variants are produced: class I (1) transcripts with their initiation site on exon 1 (P1), and class II (2) transcripts that use exon 2 as a leader (P2) [19,21,23,24]. Furthermore, research notes that GH stabilizes the more mature class II IGF1 transcripts, as well as destabilizes mature class I transcripts, indicating that class II mRNAs might have a bigger translational potential [68]. Moreover, other research results suggest that class II transcripts are probably more stable and play a mainly endocrine function [68,69]. Differences in the transcription initiation at two promoters were reported in rats and mice, with class I Ea expression in rat liver and class II Ea in mouse liver representing 90% and 70% of the total IGF1 mRNA, respectively [70].

In both *IGF1* promoters, multiple specificity protein 1 (SP1) binding elements are present, which indicates that SP1 can be an important *IGF1* transcription regulator [48]. Other study revealed that *IGF1* promoter and transcription factor Runx2, activating an upstream response element involved in *IGF1* promoter activity, were both reduced by prolonged hypoxia in cortisol treated and prostaglandin E2 (PGE_2_)-stimulated rat osteoblasts [71].

### 3.2. IGF1 Gene Alternative Splicing and Promoter Usage

On the 3′ *IGF1* terminus, as a result of AS, at least three transcripts are formed: IGF1Ea (IGF-IA; an exon 4–6 splice variant, lacking exon 5, the most abundant variant; present in humans and rodents) [26,51,72], IGF1Eb (an exon 4–5 splice variant, lacking exon 6; present in human liver; variant unique to humans), IGF1Eb (an exon 4–5–6 splice variant; present only in rodents) [4,9,73,74] and IGF1Ec (Mechano-growth factor, MGF; also identified in human liver; contains exons 4, 49 bp of exon 5, and exon 6 (exon 4–5–6) and results in translation of Ec peptide) [4,21,24,73,75]. Human Ec peptide shares 73% homology with rat Eb peptide and is considered to be its counterpart [75]. Exon 3 encodes a portion of signal peptide, as well as the mature protein common for all isoforms, while exon 4 is translated into the rest of the mature peptide and the proximal part of the E domain. On the 5′ terminus of IGF1, the presence of two promoters, as well as alternative transcription start points, result in generation of class I (1) and class II (2) mRNA isoforms [4,55,73,75,76,77].

Finally, as result of alternate promoter usage and differential splicing of *IGF1*, at least six variants (isoforms) of mature mRNAs, i.e., IA, IB, IC, IIA, IIB, and IIC are generated in humans. All transcripts contain exons 3 and 4, encoding the mature 70-kDa IGF1 protein, while exons 5 and 6, which are alternatively spliced, code for the three C-terminal E peptides: Ea (exon 6), Eb (exon 5) and Ec (fragments of exons 5 and 6) [4,21,23,54,78] (Figure 1).

The alternatively spliced *IGF1* generates multiple transcripts, each encoding a different prepro-IGF1 proteins with variable signaling peptide leader sequences [21,24]. In mammals, the multiple mRNA species range in size between 0.7 and 7.6 kb [19,20,56,80]. The human exon 5 has a nucleolar localization signal situated in the C-terminal part, encoding the Eb domain [81] and a polyadenylation site [75]. The presence of at least four polyadenylation sites in the 3′-UTR of the rodent and human IGF1 mRNAs was proven, encoded by exon 6, which increases the possibility of posttranscriptional regulation, as well as formation of different sized transcripts of *IGF1* [24]. In all of the studied animals, the IA (Ia) IGF1 mRNA isoform is predominantly produced, as it is also the most conserved across the species [76,82,83].

### 3.3. IGF1 Protein Processing

Different IGF1 mRNA variants might potentially code many types of prepro-IGF1 precursor proteins, with the specific differences in their 5′-UTR providing the molecular basis of translational control of this protein’s biosynthesis [19,23,29]. The IGF1 gene is first translated into prepro-IGF1 precursor protein composed of a signal peptide, signal peptide cleavage site, IGF1, pro-protein convertase cleavage site, and the E-peptide. The signal peptide is then removed during translation. In humans, three E-peptides are produced: EA (Ea), EB (Eb) (unique to humans), and EC (Ec) [9,12,22,84,85,86] (Figure 1B). It was proven that human Ec peptide is produced simultaneously with its precursor protein (pro-IGF1Ec) in vivo, in different types of tissues (e.g., muscle, liver, heart) [87].

Mature, biologically active IGF1 protein is formed after the posttranslational cleavage of the pro-IGF1 isoforms (pro-IGF1A, pro-IGF1B, and pro-IGF1C) and the removal of E-peptides on the 3′ end [22,83,86]. It is known that different pro-peptides might act as separate growth factors (e.g., IGF1Eb, IGF1Ec), as well as modulate the mature IGF1 activity [22,84,85,88]. It was suggested that the production of the mature IGF1 protein can be independent of the expression of isoforms themselves. A team of Elisabeth R. Barton proved that both E pro-peptides (EA and EB) are not essential for IGF1 secretion. However, both increased cell entry of IGF1 from the culture media. Hence, they modulate and possibly intensify the bioactivity of IGF1 [89]. Later, it was proved that E-peptides are characterized by low independent activity, as mitogenic effects in the muscle tissue are mediated by IGF1R [86]. In turn, functional studies of Durzynska et al., conducted in murine skeletal muscle, showed that, in comparison to mature IGF1, non-glycosylated form of pro-IGF1 has a similar ability for IGF1R activation, while its glycosylated variant significantly reduces this receptor’s activity [90]. Recent research on the mechanisms of IGF1 production proved that E-domains contain intrinsically disorders regions (IDRs), which regulate the pro-IGF1s production and secretion and are responsible for many posttranslational modifications (e.g., acetylation, glycosylation, methylation, phosphorylation, controlling protein half-life, etc.) [88]. The authors identified a highly conserved N-glycosylation site in the Ea-domain (not present in pro-IGF1Eb and pro-IGF1Ec), regulating intracellular levels of pro-IGF1Ea and preventing its degradation in proteasomes. Additionally, it was proven that both the alternative Eb- and Ec-domains control the subcellular localization of pro-IGF1s, leading to nuclear localization of both pro-IGF1Eb and IGF1Ec isoforms [88]. Previously, a nuclear and strong nucleolar localization of an isoform of the human IGF1 precursor (exon 5 containing chimeras) was described, which might suggest an autocrine function of this isoform [81]. However, in contrast to the other components of the IGF1 axis, the molecular effects of nuclear pro-IGF1Eb isoform and its derivatives are not completely known [91].

### 3.4. Regulating Mechanisms of IGF1 Gene Alternative Splicing

In humans and animals, the *IGF1* transcription is complex and controlled by different mechanical [92], hormonal [5,14,62,63,93], nutritional [64,93,94,95], and developmental factors [63,70,93,96].

It’s already certain that *IGF1* is a key target of GH activation [4,5,62,97]. In the 1990s, it was proven that GH influences the transcription of all IGF1 mRNA splicing and polyadenylation variants [98]. The insulin or GH alone increased the number of transcripts initiated from exon 1, with the effect of the combination of both hormones further promoting generation of these transcripts [62]. It was noted that GH stimulates recruitment of latent transcription factor Stat5b to multiple dispersed regions within the *IGF1* locus [99], indicating this factor as a crucial mediator of the GH-IGF1 biosynthetic pathway [6,7]. Furthermore, the total hepatic IGF1 mRNA is increased by GH, as well as protein and energy status. Increased dietary energy and protein induced expression from both class I and II transcripts, while GH treatment alone only increased the presence of class II transcripts in sheep liver. In turn, muscle IGF1 expression was about 20-fold less than in liver and consisted mainly of class I transcripts [5]. During fasting, the lack of IGF1A transcript reduction in rat liver was observed with a decrease in the numbers of steady-state mRNAs [94]. This suggests that posttranscriptional mechanisms are responsible for a decrease in stability of liver IGF1 mRNA in vivo [94,95]. It was also demonstrated that cAMP-dependent protein kinase (PKA) is implicated in the control of *IGF1* transcription by PGE_2_. In 5′-UTR of exon 1, a short segment of *IGF1* promoter was identified (termed HS3D binding site), essential for such hormonal regulation [100]. Exon 5 of *IGF1* contains a regulatory 18-nucleotide purine-rich exonic splicing enhancer (ESE) site that recruits serine-arginine protein splicing factor-2/alternate splicing factor (SRSF1), as well as increases the efficiency of exon 5 inclusion into the transcript from 6% to 35%. The SRSF1 is activated by phosphorylation or cellular localization, which may mediate tissue and hormonal specific splicing of the IGF1 transcript [29].

In addition to species-dependent *IGF1* transcription variation, there is also evidence of tissue-specific expression, as the processing of IGF1 mRNA in the liver is different than in non-hepatic tissues [5,70,76,82,93,94,101,102].

### 3.5. Phylogeny of the IGF1 Gene

IGF1 gene is evolutionary conserved in mammals and nonhuman primate species. It shows similar level of gene organization and nucleotide sequence among 25 different mammalian species representing 15 different orders and ranging over ~180 million years of evolutionary diversification [27], as well as among six nonhuman primate species from a common ancestor representing > 60 million years of evolutionary diversification [28].

Considering different variant of transcripts, it was proven that only IGF1Ea is conserved among all vertebrates, whereas IGF1Eb and IGF1Ec arose during evolution through Mammalian Interspersed Repetitive-b (MIR-b) element exonisation. This element’s appearance in the *IGF1* resulted in exon 5 gain during the evolution of mammals. AS of this exon allowed for the inclusion of new elements of regulation, at both mRNA and protein level, with the potential for mature IGF1 regulation in various tissues and species. While constitutive expression of both IGF1Eb and IGF1Ec mRNAs is present in all of the mammals, their expression rate varies greatly between different species [103].

## 4. IGF1 Isoforms and Their Function in Major Body Tissues

### 4.1. Epithelial Tissue

Expression of different IGF1 isoforms was detected in normal epithelial cells in vivo and in vitro. All three of the IGF1 mRNA isoforms (Ea, Eb, Ec) were found in cells in vivo, e.g., colonocytes [104,105], normal urothelium [33], epithelial cells of uterine cervix [49], and in vitro cell lines, such as human embryonic kidney (HEK293 cells) and human lens epithelium (HLE-B3 cells) [106]. The expression of IGF1Eb isoform and class II over class I transcripts was predominantly detected in normal colon mucosal cells [104]. Other studies showed increased IGF1Ea isoform, and class I over class II mRNAs expression in normal colon tissues [49,105].

In turn, the glandular cells of the eutopic endometrium were negative for all the IGF1 mRNA isoforms [107]. Furthermore, weak positive or negative expression of IGF1Ec (MGF) (mRNA, peptide) was observed in normal prostate epithelial cells [108,109]. Finally, the expression of mIGF1 (IGF1Ea isoform) (RNA, protein), as a potential stimulator of hair follicle morphogenesis and cycling, as well as re-epithelialization of skin wounds, was detected in keratinocytes in a transgenic mouse model [110], while human umbilical vein endothelial cells (HUVECs) and normal colon tissues were negative for MGF peptide using anti-MGF-conjugated gold-nanoparticles (AuNPs) [34].

The “locally acting” IGF1 mRNA isoform (XO6108) expression (44% of the total IGF1 mRNA) was noted in both rat hepatocytes and cholangiocytes, increasing during cell proliferation and decreasing during cell damage [111]. Similarly, the production of all three IGF1 mRNA isoforms was confirmed in normal human hepatocytes [112].

### 4.2. Connective Tissue

GH-IGF1 axis is important in the control of linear growth, as an anabolic factor for skeletal development and bone function [100,113,114,115,116,117]. IGF1 signaling is required in normal osteoblasts, especially during bone formation and remodeling [71,78,118].

IGF1 mRNA was detected in chondrocytes of the growth plate [96,117,119]. Some authors state that class I IGF1Ea mRNA plays the predominant role during the maturation of the growth plate in rats [96]. In turn, others consider the expression of so many IGF1 mRNA isoforms in rat growth plates as a proof of transcriptional differences, rather than animal’s developmental state [119].

When it comes to the expression of IGF1 isoforms in different types of connective tissue (cartilage, bone, adipose tissue) study results mostly consider the MGF isoform (mRNA, protein). In osteoblasts subjected to cyclic stretching, induced expression of MGF mRNA was described [120]. In turn, MGF peptide inhibited osteoblast differentiation and mineralization [121,122]. On the other hand, some authors report that while the MGF peptide only constitutes to around one third of all IGF1 isoforms expressed in the growth plate, it does not contribute to proliferative effects on exogenously added porcine chondrocytes [123]. Furthermore, the studies of Armakolas et al. point to the role of this peptide in human mesenchymal stem cells (hMSCs) mobilization and differentiation towards chondrocytes, enhancing the role of TGF-β1 in hyalin cartilage repair [124]. However, in C3H/He/J (C3H) mice with higher skeletal IGF1 and greater bone mass than C57/BL/6J (B6), the study showed differences in usage of proximal and distal polyadenylation site in 3′UTR, encoded by exon 6 (long IGF1 3′ UTR isoform). While total IGF1 mRNA value did not differ in the course of osteoblastic differentiation in both mice breeds, the distal polyadenylation site usage was increased in B6 mice but not in CH3 [125]. Inhibiting influence of hypoxia on the expression of IGF1 was also proven in osteoblasts, in part by limiting of an upstream Runx response element in the *IGF1* promoter [71].

Recently, there have been some attempts of using the functional role of IGF1 precursor proteins in therapy of diseases affecting bone and cartilage [126,127]. Shi et al. proved that the activity of pro-IGF1A, dependent on N-glycosylation at Asn92 in the EA peptide, determined heparin binding and caused an increase of biosynthesis in bovine articular chondrocytes. This property could be used in therapy of cartilage damage caused by an injury or osteoarthritis [126]. A recent study on the rabbit bone injury model, using 52 different synthetic MGFs, proved that the T-MGF19E peptide can significantly promote osteoblastic MC3T3-E1 cells proliferation, differentiation, and mineralization, and thus promote bone injury healing [127].

The role of MGF was also implicated in the promotion of adipogenic differentiation and migration of rat bone marrow MSCs [122].

### 4.3. Muscle Tissue

IGF1 is an important positive regulator of muscle mass, playing a mainly autocrine/paracrine role in this tissue [24,128]. Additionally, proper levels of IGF1 in muscles is not only implicated in their growth, but also in the maintenance of the glucose homeostasis. IGF1 independently modulates anabolism and muscle metabolism in an age-dependent manner [129].

#### 4.3.1. Skeletal Muscle

IGF1 mRNA isoform expression was described in both human and animal striated muscle tissue. Heterogeneity of this expression is linked to early stages of skeletal muscle development in physiology [92], muscle loss in aging (sarcopenia) [85,130], skeletal muscle stretching [92,131,132], as well as excessive exercises and/or muscle injury [20,87,118,133,134]. Differential expression of IGF1 mRNA isoforms is also regulated by other mechanisms, depending on muscle activity [135,136].

Locally, in skeletal muscle, class I (exon 4–6 spliced) variant (IGF1Ea), also known as “local muscle specific” (mIGF1) or “liver muscle specific variant” (L.IGF1), is most strongly represented [24,132,135]. In developing muscle, this isoform initiates the fusion of myoblasts to form myotubes and is constitutively expressed in both mouse myoblasts and myotubes. In turn, MGF plays a pro-proliferative role in myoblasts that are necessary for secondary myotube formation and establishment of the satellite (stem) cell pool. Furthermore, the expression of both isoforms was found to be upregulated by both a single ramp stretch and cycling loading [92]. However, in a model of atrophic muscle, it was observed that mature IGF1 protein is better in promoting muscle growth and regeneration, compared to IGF1A pro-peptide with maintained E domain [137]. Recent studies, using CRISPR-Cas9 gene activation approach, note an upregulation of different IGF1 variants in cell lines derived from mouse and human skeletal muscle myoblast, as well as a selective upregulation of class I/II IGF1 transcripts through the change of the target location of a single-guide RNA. These isoforms promoted myotube differentiation and prevented dexamethasone-induced atrophy in myotubes in vitro [128].

Local expression of two pro-peptides, IGF1Ea and IGF1Eb, plays an important protective role for age-related loss of muscle mass/force. This action includes autophagy/lysosomal system activation, increased Peroxisome Proliferator-activated Receptor Gamma Coactivator 1-alpha (PGC_1_-α) expression, modulating mitochondrial quality, reactive oxygen species (ROS) detoxification, and inflammatory state, as well as the maintenance of morphological integrity of neuromuscular junctions. It was also proven that the IGF1Ea isoform is expressed as mature IGF1, while IGF1Eb acts in an unprocessed form. Furthermore, only the IGF1Ea expression promoted a pronounced hypertrophic phenotype in young mice, which was later maintained with age [130].

The most spectacular role in skeletal and myocardial remodeling, hypertrophic adaptation, as well as muscle tissue regeneration after injury is played by IGF1Eb (IGF1Ec in human) (the exon 4–5–6 splice variant) [75,87,118,132,138]. IGF1Eb was first described in skeletal muscle, and was originally called MGF, as its RNA is expressed in muscle in response to overload and mechanical damage [20,132,135]. It should be noted that the name MGF refers to the IGF1Eb (IGF1Ec in human) mRNA isoform, a 24 aa COOH-terminal peptide separated from the IGF1, as well as synthetically manufactured peptides [20,139,140]. It was shown that, after muscle injury, a quick and transitional up-regulation of MGF mRNA (IGF1Ec isoform) expression can be observed, followed by prolonged increase in the expression of the remaining isoforms [141]. In turn, an increase in IGF1Eb (IGF1Ec in human) mRNA production correlates with markers of satellite cells and myoblast proliferation, while the upregulation of IGF1Ea mRNA is linked with mature myofiber differentiation [20].

Studies of the function of the MGF peptide in vitro showed that it enhances myoblasts proliferation [92,135,142,143], inhibits progenitor/satellite cell differentiation (facilitating repair) [143,144], activates satellite cells after local tissue damage [92,145], promotes secondary myotube formation or the fusion of myogenic cells [92,138,142], and/or enhances migration of myogenic cells through modulation of the fibrinolytic and metalloproteinase systems [146].

A direct and preferential stimulating influence of GH on MGF prior to IGF1Ea expression was described, potentially leading to the activation of muscle satellite cells [147]. Furthermore, relations between the activation of MGF expression and cAMP level in the myoblasts [148], as well as the activation of IGF1Ea and MGF (mRNA, protein) via myofibrillar proteins (sub-fragments of titin and myomesin) released from the damaged muscle were also shown [148,149]. In vivo mouse model studies showed that both E-peptides’ (EA and EB) expression altered Extracellular Signal-regulated Kinases 1/2 (ERK1/2) and Akt phosphorylation, as well as increased satellite cell proliferation, while EB alone was more potent at increasing muscle hypertrophy [150].

Some of the in vitro studies question the pro-proliferative role of the MGF peptide in human skeletal muscle myoblasts and primary mouse skeletal muscle stem cells, in the same time showing a proliferative effect of mature IGF1 or full-length IGF1Eb isoform administration in both cell types [140].

In vivo studies of muscle response to eccentric contraction proved an age-dependent decrease of MGF production in humans [135], rats [151], and mice [136]. Furthermore, they confirmed the age dependent differences in IGF1Ea and IGF1Eb isoform action, suggesting that the bioavailability of IGF1Eb isoform diminishes with age [136]. In humans, following muscle damage, it was shown that temporal response of MGF is probably related to the activation/proliferation phase of the myogenic program, while IGF1Ea and Eb may be temporally related to differentiation [152].

#### 4.3.2. Cardiac Muscle

The influence of different IGF1 isoforms is also studied in different models of cardiac muscle repair (mostly myocardial infarction related) [153,154,155,156,157]. An increase of IGF1Ea and MGF isoform expression (mRNA, protein) during the late postinfarction period was observed in rat myocardium [153]. A cardioprotective role of mIGF1 in murine cardiomyocytes was also reported, preventing oxidative and hypertrophic stress via NAD(+)-dependent SirT1 deacetylase [154]. Other mouse model research noted a cardiomodulating and cardioprotective influence of a synthetic analog of the MGE peptide, administered during myocardial infarction and affecting local production of mature IGF1 [155]. Other studies confirmed this observation, with the human MGF peptide administration to mice facilitating the actions of the locally produced IGF1 during the progression of heart failure to improve cardiovascular function [156]. In mice injected with the MGF peptide eluting microrod scaffolds decreased mortality, ameliorated decline in hemodynamics, and delayed decompensation were observed [158].

### 4.4. Nervous Tissue

IGF1 itself influences brain development, myelinization of nerve fibres, as well as functions of the adult brain (reviewed in [159]). Additionally, it provides neuroprotection to oligodendrocyte progenitor cells and improves neurological functions following cerebral hypoxia-ischemia in the neonatal rat [160]. Some studies showed that MGF protected facial neurons after nerve damage [161] and had strong neuroprotective effects against ischemia in a gerbil model of transient brain ischemia in vivo and in vitro [162]. Furthermore, in mouse brain, MGF promotes neurogenesis and increases the number of neural progenitor cells, preventing neuronal attrition and brain dysfunction in aging mice. In turn, it does not affect adult new-born neurons at post-mitotic stages [163]. A detailed mechanism of the neuroprotective activity of an MGF derivative via Protein Kinase Cϵ (PKCϵ) activity and Nuclear Factor (NF) E2-related factor 2 (Nrf2) was also described [164]. Finally, IGF1 short peptides are characterized by good blood–brain barrier penetrance, which may also serve for therapeutic purposes [159].

## 5. IGF1 Gene Alternative Splicing and Carcinogenesis

As mentioned, in several tissues of mammalian species and in vitro cultured cells, IGF1Ea isoform constitutes a majority (>90%) of all the expressed IGF1 mRNA isoforms. In physiology, expression of this IGF1 isoform is mainly located in the liver, skeletal muscles, and adipose tissue [76,103].

Some human tumors also dominantly produce the IGF1Ea isoform, e.g., colorectal cancer (CRC) [104,105] and cervical cancer (CC) [49]. A majority expression of this isoform was also confirmed in cultured CC [31], breast cancer (BC), endometrial cancer (EC), myelogenous leukemia, and melanoma cells [31,106].

In other human cancers, an increase in expression of the two other mRNA isoforms, IGF1Eb and IGF1Ec, as well as respective IGF1 pro-peptides might also be observed [30,106,165].

Among the known IGF1 mRNA variants and IGF1 precursor proteins, a clearly mitogenic role in tumor biology is attributed to the IGF1Ec isoform [30,32,108,109,166]. A confirmation of this statement can be found in studies of human cancer cell lines [14,16,106,167]. In turn, the role of the IGF1Eb isoform (anti-cancer or cancer promoting?) remains less known [31,90,168].

Carcinogenesis is connected to the higher affinity of certain IGF1 variants to IGF1R [2,118], as well as the effects of IGF1 isoform action independent of IGF1R, INSR, and IGF1/INSR [14,84,107,108,166,168].

### 5.1. IGF1 Isoform Expression (mRNAs, Proteins) in Selected Tumors

In human cancers, the expression of numerous IGF1 mRNA isoforms (Ea, Eb, Ec, class I and II transcripts, pro-peptides, peptides) was studied, both in in vivo tumor tissues and in a variety of animal models or cultured human/animal transformed cells. In in vivo studies of cancer samples, either small-scale differences were indicated between the expression of class I and class II transcripts (e.g., CRC) [104,105], or the majority of class I over class II was described (e.g., CC) [49]. In cultured neoplastic cells, class I proved to be the predominant origin of IGF1 transcripts from most cell lines, especially for IGF1Eb and IGF1Ec isoforms [106].

#### 5.1.1. Breast Cancer (BC)

In the case of BC, the expression and biological role of IGF1 isoforms (mRNAs, proteins) was only studied in in vitro models [16,106,167,168,169,170].

Research on MSF7 (estrogen-, progesterone-, and glucocorticoid receptor-positive breast adenocarcinoma) and MDA-MB-231 cells (negative for the mentioned receptors) showed the expression (using the fold change formula, as opposed to relative expression of these isoforms) of IGF1Ea (exon 4 sense; exon 4–6 anti-sense) and IGF1Ec isoform corresponding to the last 24 aa of bioactive protein (Ec peptide) (exon 5 sense, exon 6 anti-sense isoforms). In the same time, no expression of IGF1Eb (exon 4 sense; exon 5 anti-sense) was detected in both of these cell lines, with the expression of the Ec peptide in MCF7 cells described as the lowest among all 12 studied cell lines. One of the lowest levels of Ec peptide expression might indicate the role of other factors (apart from cancer type and hormone receptor status) in a given pattern of IGF1 mRNA expression in BC [106].

The results of studies by Chen et al. on the role and effects of recombinant rainbow trout (rt) pro-IGF1-E-peptides’ action in human cancer cell lines, indicated anticancer activities of the recombinant rtEa4-peptide [human Eb peptide (hEB) analogue] on cultured human BC cell lines (MCF7, MDA-MB-231, ZR751 cells) [168,169]. The E peptide induced morphological changes in cells, promoting sensitivity to drugs like alpha-amanitin or cycloheximide, reducing colony formation activity in MDA-MB-231 cells and enhancing cell attachment [169]. It was noted that this peptide cooperates with alpha2/beta1 integrin receptors at the cell membrane, inducing the expression of fibronectin 1 and laminin receptors genes [171]. It was also demonstrated that inhibition of the MDA-MB-231 cell invasion using the rtEa4-peptide was mediated via the suppression of Urokinase-type Plasminogen Activator (uPA) and tissue-type PA and PA Inhibitor 1 (PAI1) gene activities. Both forms of the rtEa4 peptide (hEb peptide) activity occur through classical signaling pathways: PI3K, phosphokinase C (PKC), Mek1/2, JNK 1/2, and p38 MAPK [171,172]. The group discovered that hEb suppresses growth and tumor-induced angiogenesis, seeding MDA-MB-231 cells on the chorioallantoic membrane of 5 days old chicken [168]. Further studies allowed for production and functional characterization of recombined hEb. It was proved that the peptide has structural antimicrobial peptide (AMP)-like characteristics, which binds to the MDA-MB-231 cell plasma membrane and interacts with hydrophobic tissue culture substratum. The hEb enters BC cells via clathrin-mediated endocytosis, which is a key player for cell lamellipodia outspread [173]. Recently the mentioned authors isolated receptors/binding components (eight protein molecules bound reversible with hEb) of this peptide from MDA-MB-231 cells. These protein components are located in the plasma membrane (e.g., GRP78, ANXA2), while others localize to different subcellular compartments. A study of the hEb peptide in BC, in vivo and in vitro, supports the hypothesis of IGF1R-independent hEb peptide action and biological activity against cancer cells [167].

Other studies uncovered pro-proliferative properties, causing induced growth of BC cells (MCF7, T47D and ZR751), of all IGF1Ea, IGF1Eb, and IGF1Ec pro-forms, as well as showed that this effect is independent of the mature IGF1. Additionally, it was reported that IGF1 pro-forms are capable of IGF1R phosphorylation, in an extent lesser than the mature peptide. Anti-IGF1R neutralizing antibodies completely inhibited the action of IGF1 pro-forms, while anti-IGF1 antibodies inhibited the mature peptide, but only partially inhibited the biological activity of the IGF pro-forms [16]. Other study showed that the application of synthetic human Ec (hEc) peptide stimulated proliferation of MCF7 cells, but not hormone-resistant MDA-MB-231 cells. The MCF7 cells with stable expression of hEc have a greater proliferation and migratory capacity in comparison with mock or wt-type cells, as well as enhanced activity of the intracellular ERK1/2 pathway [170].

Summarizing, in both types of BC cells (estrogen receptor (ER)-positive and ER-negative) the expression of the IGF1Ea and IGF1Ec isoforms was reported, with total absence of IGF1Eb transcripts. It was also shown that the rtEa4-peptide and its counterpart, the hEb peptide, exhibits anti-cancer activity in BC cells through, e.g., inhibition of growth, invasion of BC cells, and cancer-related angiogenesis. This action is independent of IGF1R, with identification of new hEB receptors/binding spots on BC cells possibly allowing to better determine the role of this IGF1 isoform in BC. In turn, in the case of the hEc peptide, pro-proliferative activity seems to depend on the phenotype of BC cells (only observable in hormone-sensitive BC cells). IGF1 pro-forms also intensify the proliferation of BC cells, as well as exhibit the ability for IGF1R phosphorylation.

#### 5.1.2. Colorectal Cancer (CRC)

Potential correlation between the levels of circulating IGF1 and CRC risk were reported in recent literature [174]. Similarly, lowered IGF1 protein expression was noted in cancer tissue compared with normal mucosa [175].

In vivo studies on the role of different IGF1 transcripts/pro-peptides expression in CRC can be found in singular publications [34,104,105].

Our first research on the expression of different IGF1 transcripts in CRC showed quantitative dominance of the IGF1Ea isoform (A), followed by isoform B and C in CRC tissues. In turn, in control large intestine, the IGF1Eb mRNA exhibited the highest expression [104]. In a study conducted on a bigger study group an even larger percentage of all the analyzed isoforms was represented by the IGF1Ea mRNA (A) over B and C variants (82%, 17%, and ~1%, respectively). While quantitative evaluation failed to confirm any significant differences between the expression of A and B isoforms, both have been significantly more abundant than C isoform. Similarly, in control large intestine, the quantitative analysis also has failed to detect significant differences between the expression of A and B isoforms. The expression of isoform C was found at the lowest level, representing about ~1% of all isoforms analyzed in various types of tissues. No quantitative differences have been detected in mRNA expression of the two promoter types (class I and II transcripts), suggesting that genomic expression of *IGF1* remains under a control of both P1 and P2 promoters in CRC and in normal colon. The quantitative analysis showed significantly lower expression of the total IGF1 mRNA, similarly to the results described by other authors [175]. Expression of all IGF1 transcripts in CRC was also lower compared to control samples [105].

Furthermore, our own studies on the expression of IGF1 transcripts in pre-cancerous lesions of the colon (ulcerative colitis, Crohn’s disease, adenoma, hyperplastic polyps) showed a majority of IGF1Eb (B) isoform expression (72%), over A (26%) and C isoforms (2%). Quantitative analysis confirmed the dominant expression of IGF1Eb mRNA isoform, but significant differences were only reported between the expression of the IGF1Ea (A) (more) and IGF1Ec (C), as well as B (more) and C isoforms. There were no differences in quantitative expression of P1 (class I) and P2 (class II) transcripts. Strong positive correlation was indicated between the expression of IGF1Eb mRNA and Proliferating Cell Nuclear Antigen (PCNA). Additionally, higher expression of IGF1Eb mRNA was noted in precancerous lesions than in CRC. While, based on these observations, it may seem that the IGF1Eb mRNA isoform may play an important role at the early stages of human colon carcinogenesis, the mechanisms of its action in CRC caused by chronic colonic inflammation are not yet known [176].

Other studies confirmed the expression of the MGF peptide in CRC, colonic polyps, as well as large intestine polyps that were not present in normal colon. Anti-MGF-conjugated AuNPs were localized in epithelial cells [34]. However, no expression of the MGF was determined in colonic and retrosigmoidal neuroendocrine neoplasms (NENs) [32].

Chen et al., in their in vitro studies, showed anticancer activities, as well as some mechanisms of recombinant rtEa4-peptide (hEb peptide) action on CRC cells (HT-29 cells) (e.g., inhibition of anchorage-independent growth) [169]. Other in vitro studies in human colon adenocarcinoma DLD1 cells revealed that the IGF1Ea and IGF1Eb isoforms expression was similar. Both isoforms were expressed in high levels (using the fold change formula), as compared with other cell lines examined in the study. In turn, the IGF1Ec isoform (Ec peptide) expression was rather low. Comparable expression of class I and class II transcripts was observed [106]. Positive MGF peptide expression was also confirmed in two other CRC cell lines (SW620 and HT29) with uniform fluorescence [34].

In summary, in in vivo and cultured cell studies of colorectal cancer, expression of all of the IGF1 mRNA isoforms was confirmed, with advantage of isoform IGF1Ea over IGF1Eb and IGF1Ec when it comes to percentage, but similar quantitative expression of both IGF1Ea and IGF1Eb isoforms. The lowest expression was attributed to the IGF1Ec isoform. The expression of IGF1Eb isoform was the highest in precancerous lesions of the colon in vivo, positively correlating with PCNA. Independently of the study model, a comparable presence of class I and II transcripts was confirmed, enabling the production of all of the IGF1 mRNA isoforms both in vivo and in vitro. In vivo, CRC showed decreased expression of total IGF1 mRNA and all of IGF1 isoforms compared to control. The results cannot be related to in vitro studies, as no relative expression analysis was performed, and control consisted of cell lines of different origin than the colon. While anticancer activity of the rtEa4-peptide (hEb peptide) was described in cultured CRC cells, estimation of the mechanisms of IGF1Eb action in CRC in vivo requires further studies.

#### 5.1.3. Endometrial Cancer (EC)

In this type of cancer, the studies on the role of IGF1 mRNA isoforms only concern endometrial cancer cells in vitro (KLE) [106,107]. Milingos et al. showed expression of all three IGF1 mRNA isoforms in the stromal cells of the eutopic and ectopic endometrium, while only the IGF1Ec isoform (mRNA, protein) was detected in glandular cells of the ectopic endometrium. The authors also noted that the MGF peptide stimulates the growth of the KLE cells in a IGF1R- and INSR-independent manner [107]. Other studies on KLE cells showed the highest expression of Ea mRNA, as well as mRNA encoding the Ec peptide, with the expression of Eb comparable to other cell lines. Presence of all the IGF1 variants was described, belonging to both class I and class II IGF1 transcripts. Interestingly class II transcripts production was the highest in the KLE cells (among the studied cells lines), particularly in relation to the IGF1Ea isoform and IGF1Ec (Ec peptide) [106]. Furthermore, it is worth reminding that class II transcripts are probably more stable, and have a mainly endocrine function [68,69].

#### 5.1.4. Epithelial Cervical Cancer (CC)

The oncogenic types of Human Papillomavirus (HPV) cause the vast majority of CC [177,178]. The interactions between HPV and all IGF axis components (including IGF1) in CC development are already presented in excellent reviews [165,179].

In quantitative expression analysis of different IGF1 mRNA isoforms in various stages of CC in vivo, overexpression of all of the IGF1 mRNA isoforms (Ea, Eb and Ec) was detected in precancerous lesions (L-SIL, H-SIL) (HPV-positive), as compared to control (HPV-negative) and cervical cancer (HPV-positive). In CC, by percentage, the highest expression was attributed to IGF1Ea (85%) vs. Eb (14%) and Ec (1%). The percentage of Eb in total IGF1 was higher in CC (14%) than in other stages of carcinogenesis (8–10%). In around 70% of studied CC tissues, P1 promoter transcription (class I transcripts) was detected. However, in the same time, the participation of P2 activity was the highest in cancer (31%) compared to other tissues (precancerous lesions, control cervix). A significant correlation was also detected between the expression of FOX2 AS regulator (RNA binding motif protein 9; RMB9) and the expression of IGF1Eb in CC and precancerous lesions. Positive correlation between the expression of FOX2 mRNA and IGF1Ec concerned precancerous lesions only (L-SIL, H-SIL). When it comes to the overexpression of transcription factor SP1, it was proven that it positively correlates with IGF1 upregulation only in precancerous lesions, with no correlations with P1/P2 activity detected in CC tissue [49]. Furthermore, positive cytoplasmic expression of the MGF was also detected in one study concerning neuroendocrine uterine cervical cancer [32].

In vitro studies of IGF1 mRNA isoforms expression in cultured CC cells transformed with HPV18 (HeLa cells), showed the highest expression of the IGF1Ea isoform (relative copy number (c/n) of 5.67), detected only in the cytoplasmic fraction. In turn, IGF1Ec transcript copy number was very low (0.1). The highest levels of pro-IGF1Ea levels were also detected in HeLa cell extracts (including the cytoplasmic fraction), as compared with other cell lines. However, no hEa peptide (~4 kD) was noted. With long exposure times, the anti-hEb antibody immunoblotting revealed a ~9 kD band (from mature IGF1), with a ~16 kD band also evident, corresponding to the predicted size of the pro-IGF1Eb isoform. Furthermore, the human Eb peptide was present only in the nuclear fractions. In HeLa cells, it was noted that the levels of transcript expression (lower) do no always correspond to the levels of the protein (higher content), especially its glycosylated form. This mostly concerns the pro-IGF1Ea protein [31]. According to the authors, cultured cell studies should also take the role of the HPV and HPV oncogenes (E6 and E7) into account, as they might influence the spliceosome elements directly, or indirectly via the activation of splicing factors [179]. Earlier studies of this group concerning the potential motogenic and mitogenic bioactivity of hEb, showed that synthetic hEb enhances cell growth of HeLa cells. Moreover, a mainly nucleolar/nuclear localization of this peptide was confirmed in cancer cells [90]. Other study on HeLa cells showed a quantitatively similar expression of IGF1Ea and IGF1Eb isoforms. Interestingly the IGF1Ea isoform was produced exclusively from class II mRNAs, while Eb and the Ec only derived from class I transcripts. Furthermore, the expression of the Ec peptide was characterized by levels higher compared to other studied cell lines. However, the study did not compare the relative expression of IGF1 isoforms’ expression to control group and between groups, which limits this study’s comparability and interpretation [106].

Summarizing, in CC in vivo, an increase of expression of all IGF1 isoforms was observed in precancerous lesions, compared with CC and control. However, the shift in the balance between IGF1 isoforms towards IGF1Eb in CC might play a role in cancer development and progression. In in vitro conditions, some authors reported comparable expression of IGF1Ea and IGF1Eb, while others noted bigger presence of IGF1Ea than other isoforms. Both in vivo and in vitro, differential participation of both IGF1 gene promoters was described, leading to formation of class I and class II transcripts (with a majority of class I transcripts). Alternative splicing factors (e.g., FOX2), as well as possibly HPV proteins, participate in regulation of the *IGF1* promoter activity and formation of different transcription patterns. It is said that the stability of the pro-IGF1Ea protein in CC cells might also be influenced by HPV oncogenic proteins. Furthermore, the differences in the levels of transcripts (lower content) and pro-peptides produced (higher content), as well as mechanisms of their occurrence, are constantly being researched. In this type of cancer, the hEb peptide shows a pro-proliferative activity, while its nucleolar/nuclear localization points to a more complex role of this IGF1 isoform in CC carcinogenesis.

#### 5.1.5. Hepatocellular Carcinoma (HCC)

The liver is the main source of IGF1 protein in physiology, with this peptide involved in the development and progression of the HCC [180]. Nevertheless, singular studies concern the expression of IGF1 isoforms in different models of hepatocarcinogenesis in vivo [112,181].

As known, hepatitis C virus (HCV) is the major causative agent of HCC. Own studies on the tissue material from patients of different advancement of inflammatory changes (grading), as well as different stages of fibrosis (staging) in chronic hepatitis C (CH-C), showed a higher relative expression of IGF1Ea and IGF1Ec isoforms, class I transcripts, and total IGF1 mRNA in the CH-C as compared to the control. An increase in grading was associated with decreased IGF1 mRNA expression, an altered profile of mRNA IGF1 isoforms (lower expression of Ea and Eb transcripts), and increased expression of IGF1R mRNA. Hence, we have suggested that HCV can alter the IGF1 splicing profile [112].

There are also results of studies on diethylnitrozamine-induced HCC in mature mice, in which a lowered expression of mRNA of two isoforms (local MGF and endocrine form of IGF1) in HCC, as well as increased expression in tissue surrounding the tumor, compared to control tissues, was described. At the same time, the proportion of isoform transcripts was stable [181].

Chen et al., in their in vitro studies, showed anticancer activities of the rtEa4-peptide (hEb peptide), as well as some of the mechanisms of action of this IGF1 isoform in well differentiated human HCC cell line (HepG2 cells) (e.g., inhibition of anchorage-independent growth) [169].

Another study on IGF1 isoform expression in HepG2 cells, showed the overall IGF1 expression level lower than in K562 cells. IGF1Eb isoform exhibited the most pronounced expression in these cells, with a switch between IGF1Eb (69.9 ± 28.6 relative c/n) and IGF1Ea (28.3 ± 6.7) observed. Furthermore, IGF1Ec transcript copy number was very low. Immunoblotting demonstrated bands of 11–12 kD which could correspond to human glycosylated pro-IGF1Ea (pro-IGF1A), containing one N-glycosylation site. Finally, hEB peptide levels were half of those detected in other cells (K562) [31].

In turn, studies of Christopoulos et al., conducted on other well differentiated hepatocyte-derived carcinoma cell line (HuH7 cells), showed a quantitatively comparable expression of IGF1Ea and IGF1Eb, at levels relatively low compared to other cell lines. The IGF1Ea isoform was produced from both class I and class II mRNAs, while IGFEb and IGF1Ec only originated from class I transcripts [106].

Summarising, it is hard to compare the limited results on the role of IGF1 isoforms expression in human HCC. The hEb peptide of pro-IGF1 seems to exhibit anticancer activity in HCC cells in vitro. However, determination of the diagnostic-prognostic role of different IGF1 isoforms in human HCC in vivo requires further studies. The profile of liver expression of IGF1 isoforms is species-specific (mice/human) and most likely depends on many factors, such as acute/chronic inflammation and advanced liver fibrosis, as well as hepatotropic virus infection (e.g., HCV). Finally, full determination of the molecular mechanisms responsible for the biological activity of IGF1 isoforms in hepatocarcinogenesis also requires further research.

#### 5.1.6. Lung Cancer (LC)

Bioinformatic analysis performed on almost a thousand LC patients shows that overexpression of IGF1 correlates with high risk of cancer progression [182]. Positive cytoplasmic expression of the IGF1Ec peptide (MGF) was described in half of the neuroendocrine cancers of the lung [32]. Other studies on IGF1 mRNA/protein isoforms were only undertaken in in vitro models. Promotion of normal and malignant human bronchial epithelial cell growth was examined (NCI-H345) using 22 aa residue long synthetic analog of the E peptide of the IGF1Eb (IGF-IB) precursor (Y-23-R-NH_2_). The proliferative effect was not inhibited by the addition of a monoclonal antibody antagonist to the IGF1R alpha (alpha IR3) [84]. Research on other cell line (A549) with the use of the fold change formula showed the biggest expression of the Ea and the Ec isoforms, followed by the Eb isoform. The IGF1Ea and IGF1Ec were produced from both class I and II transcripts, while IGF1Eb isoform only originated from class I mRNA [106].

#### 5.1.7. Osteosarcoma

In osteosarcoma, the expression of different IGF1 isoforms (mRNA, protein) is only reported in in vitro model based studies [14,15,31,106].

Phillipou et al., analyzing human osteoblast-like MG63 osteosarcoma cells, detected exclusively IGF1Ea and IGF1Ec transcripts. Only the exposure to dihydrotestosterone (100 nM for 72 h) induced the transcription of Eb isoform and increased the expression of Ea and Ec. In turn, synthetic E peptide (C-terminal of the IGF1Ec isoform) stimulated the growth of MG63 cells, as did IGF1 and insulin. Whereas only E peptide stimulated the growth of IGF1R knock-out (KO) and insulin receptor (IR) KO MG63 cells, it also exerted mitogenic activity on MG63 cells via an IGF1R-, INSR-, and IGF1R/INSR-independent mechanisms [14].

Studies of IGF1 isoforms expression on other human bone osteosarcoma cells (U2OS) showed the lowest overall production of IGF1, with Eb as predominant isoform compared to other cell lines (HepG2, HeLa, K562). Furthermore, a switch between Ea (0.56) and Eb (2.62) transcript copy number was observed. With long exposure times, Eb peptide could be detected using immunoblotting [31]. Synthetic hEb enhances cell growth and increases motile properties of stable U2OS cells. A mostly nucleolar/nuclear location of hEb peptide was confirmed [90].

In turn, while studies of Christopoulos et al. on MG63 cells did not examine the relative expression, their results reported average expression of Ea and Eb compared to other cell lines, also those characterized with higher expression of Eb than Ea. Interestingly the Eb isoform was produced exclusively from class II mRNAs, while Ea and Ec only derived from class I transcripts. The expression of the Ec peptide was characterized by a relatively high levels compared to other cancer cell lines. However, the sole use of the fold change formula makes comparative analysis significantly more difficult [106].

Comparative studies of IGF1Ec (MGF) isoform expression on several human osteosarcoma cell lines (Hos, MHos and MG63) showed higher MGF mRNA expression in MG63 cells than in the MHos cell line. Administration of MGF-E peptide to MG63 cells resulted in an increase in proliferation via cell cycle progression and enhanced cell migration. The increased level of cyclin D1, CD147, Matrix Metalloproteinase 9 (MMP-9) and vascular endothelial growth factor (VEGF), and suppression of caspase 3 in MG63 cells, compared to the control group, was observed [15].

In summary, available studies indicate differential expression of all IGF1 variants, depending on the type of cultured osteosarcoma cells. Furthermore, promoting activity in this cancer type seems to be attributed to both IGF1Eb and IGF1Ec isoforms, with both of them stimulating cancer cell growth. In turn, the mechanism of mitogenic activity of the synthetic Ec peptide was independent from IGF1R, INSR and IGF1R/INSR.

#### 5.1.8. Prostate Cancer (PC)

Spares, but nevertheless valuable studies analyses the role of expression of different IGF1 variants in PC in vivo, with the use of methods such as immunohistochemistry (IHC), RT-PCR, and Western-blot analysis [108,109,166]. In PC tissues and in prostatic intraepithelial neoplasia (PIN) an overexpression of the IGF1Ec isoform (MGF) was detected. In these lesions, the expression of MGF was higher than in normal prostate tissues [108]. Mild to strong cytoplasmic expression of MGF, higher in locally advanced tumors (stage > III), was observed by other authors [109]. Similar trends were confirmed in biopsies from 78 patients with PC, showing correlation in expression of the IGF1Ec (Ec peptide) with tumor stage. Lower expression of Ec peptide was shown in PC patients with stage ≤ IIb, as compared with the tissues of patients with stage III/IV [166].

In an in vitro study model, Armakolas et al. reported preferential expression of the IGF1Ec peptide (MGF) in human androgen negative (PC3) and sensitive (lnCaP) PC cells compared to the lack of expression in normal prostate cells (HPrEC). Administration of exogenous MGF peptide promoted PC cell growth via activation of ERK1/2 phosphorylation, without affecting Akt phosphorylation. Inhibition of IGF1R and INSR silencing did not impair the mitogenic activity of MGF [108].

More recent in vitro studies on PC cells overexpressing Ec (PEc) gave additional proof for the autocrine/paracrine action of PEc, leading to epithelial to mesenchymal transition (EMT). Orthotropic injection of PEc-overexpressing HPrEC cells in SCID mice was associated with metastasis. According to authors, while PEc induces EMT through the activation of ERK1/2 pathway and ZEB1 expression via IGF1R, it can also act in a mechanism independent from IGF1R [166]. The same research group confirmed stimulation of human PC3 cells growth through the addition of synthetic human Ec peptide (hEc) to the culture, with a lack of similar effects on mouse C2C12 myoblasts. This activity was inhibited by sole addition of anti-hEc peptide antibodies, but not using a neutralizing anti-IGF1R antibody. Additionally, it was proven that hEc’s active core is located in the last four aa of its terminal end [18]. Furthermore, activating influence of interleukin 6 (IL-6) on IGF1Ec isoform up-regulation and secretion of PEc by PC cells was reported. The authors conclude that, apart from its oncogenic role, PEc participates in mobilization of MSCs, resulting in tumor repair [17].

The study of Christopoulos et al. on lnCaP cells showed one of the highest expressions of IGF1Ea and IGF1Eb among the studied cell lines, as well as top-rated expression of the Ec peptide. As the fold change formula has been used, instead of relative isoforms expression, full comparability of these results is relatively difficult. All isoforms originating from class I transcripts were the majority of total IGF1 isoforms detected. Nevertheless, IGF1Ea was also produced from class II transcripts. Additionally, studies on PC3 cells showed regulation (up and down) of pro-peptides Ea and Eb by estradiol (E2), dexamethasone (dexa) and GH treatment, in dose- and time-dependent manner. The IGF1Ec (Ec peptide) expression was significantly reduced in almost all hormonal conditions [106].

Summarising, in PC and the precancerous lesions (PIN) in vivo, a preferential expression of the IGF1Ec isoform was reported. In vitro studies also confirmed the expression of other IGF1 isoforms (IGF1Ea, IGF1Eb) in different cell lines, as well as described one of the highest levels of Ec peptide expression among the studied cell lines. Functional studies indicate that the Ec peptide (MGF) serves the role of an oncogene in PC (induced cellular proliferation, cell growth, progression through EMT, increased metastatic rate), but is also implicated in tumor repair. The mechanism of Ec peptide action usually bases on IGF1R and ERK1/ERK2 pathway (without Akt phosphorylation), but is also able to work in an IGF1R-independent manner.

#### 5.1.9. Other Cancers

In the 1990s, the expression of two IGF1 mRNA isoforms (Ea and Eb) was detected in primary human brain tumors (gliomas and the esthesioneuroblastoma) [183].

In turn, in different types of thyroid cancer (TC), sole investigation of IGF1Ec expression (mRNA, peptide), showed cytoplasmic expression of Ec peptide only in papillary differentiated TC, associated with TNM staging and capsule invasion. Higher expression of IGF1Ec mRNA was observed in more aggressive vs. non-aggressive papillary TC [35].

In urinary bladder cancer, carcinoma in situ and normal urothelium expression of all three IGF1 mRNAs has been observed (Ea, Eb i Ec). Significant downregulation of IGF1Ec isoform expression was reported in cancer, compared to normal urothelium, while increased expression of all IGF1 mRNAs were observed in in situ carcinomas. Correlation of IGF1Ec expression with clinicopathological data (tumor stage, grade, disease recurrence) was also noted. Hence, the authors suggest other role of IGF1Ec isoform, as well as the potential Ec product, in biology of this cancer [33].

Research analyzing exclusively the expression of the IGF1Ec peptide (MGF) in gastrointestinal tract NENs, i.e., gastric cancers (n = 8), pancreatic cancers (n = 17), small bowel cancers (n = 9), few cancers of unknown primary origin and appendiceal or gallbladder cancers (n = 8), report varying percentage of MGF expression (0–100% positive cases), localized solely to the tumor cell cytoplasm [32].

In in vitro studies on human immortalized myelogenous leukemia cell line (K562), Durzynska et al. showed the highest overall relative IGF1 expression, the highest IGF1Ea and IGF1Ec isoforms compared to other lines. Immunoblotting also revealed a higher Eb protein content in cell lysates compared to other cells. Additionally, the hEb peptide was present in the nuclear fraction and was the prominent in all examined cells, with no presence of pro-IGF1Eb detected. Regulatory mechanisms of the production of the IGF1Eb isoform, and by extension hEb peptide in different types of cells, still require further elucidation. In K562 cells, it was also proven that the scale of transcript expression (higher) does not always correspond to the protein levels (lower content), especially concerning glycosylated proteins, which in this case mostly concerns the pro-IGF1Ea protein expression [31].

In vitro studies of human melanoma SK-MEL28 cells, describe the highest expression of IGF1Ea, slightly lower levels of Eb (both of those proteins characterized by one of the highest expression levels among all of the studied cell lines), as well as the expression of Ec comparable to other cancer cells. All of the IGF1 isoforms originated from class I transcripts, while exon 2 served as the transcription site only for Ea isoform. However, the study did not include a comparison of the relative expression of the IGF1 isoforms in melanoma cells vs. control and between different cell lines [106].

The results of Chen et al. demonstrated biological activities of rtEa4-peptide (hEb) on human neuroblastoma cells (SK-N-F1 cells) [169,184]. It was shown that the activity of rtEa4 is similar to synthetic hEb, but not the hEa peptide. The use of this isoform in the culture of SK-N-F1 cells caused inhibition of anchorage independent growth and morphological cell differentiation with axon elongation and expression of markers typical for neural cell differentiation (e.g., neuronal-specific MAP-2, Tau, or NPY). The main pathways regulating the growth and differentiation of neuroblastoma cells via rtEa4-peptide were ERK1/2 and MAPK signaling [184].

In rat pituitary tissues, as well as cell lines derived from rat pituitary glands (GH3), both class A and class C 5′UT variant mRNAs were detected. When it comes to IGF1 isoforms, 65% of IGF1Ea and 35% of IGF1Eb mRNA was observed. All transcripts were hormonally modulated by triiodothyronine (T3) and GH [185].

Comparison of expression and role of different IGF1 variants in cancer tissues in vivo in humans and animal models, as well as in vitro is presented in Table 1 and Table 2. 

## 6. Aberrant IGF1 Isoform Expression (mRNAs, Proteins) and Cancer Therapy–Possible or Impossible Implications?

There are multiple preclinical and clinical trials of mono- and combined-IGF-targeted agent application in human solid tumors. These mostly include: anti-IGF1R monoclonal antibodies, anti-IGF1/IGF2 ligand antibodies, small molecule tyrosine kinase inhibitors (TKIs) (which bind to the receptor tyrosine kinase domain and block signaling downstream of IGF1R and INSR), IGF binding proteins (IGFBPs), IGFBP proteases, and IGFBP-related proteins (IGFBP-rPs) [186,187,188,189].

Real and potential prognostic-diagnostic importance of IGF1 isoforms (mRNA, protein) in selected human cancers is already described. However, use of this knowledge in cancer therapy is, as for now, only postulated. These available propositions are based mostly on the results of studies on the expression of different IGF1 variants, as well as the influence of recombined analogues or synthetic IGF1 isoform products on cultured cancer cells [14,84,90,168,169,171,172].

Studies of tissue expression indicate distinct roles of IGF1 isoforms in human cancer depending on the type of malignancy and different status of hormone receptors (e.g., estrogen, progesterone, glucocorticoid, androgen) on cancer cells [106,108]. Differential expression profile of the IGF1 splice variants is described in cancerous/precancerous tissues (higher) vs. normal (lower) in PC [30,108], CC [49], and HCC [181]. There are also reports of lower expression in cancerous tissue vs. control samples in CRC [104,105], or lower expression in cancerous tissue vs. precancerous alterations in CC [49].

In many cancers, the studies of expression and mechanisms of action of IGF1 splice isoforms are only based on in vitro models (e.g., breast cancer, endometrial cancer, osteosarcoma, myelogenous leukemia, or melanoma) [14,15,16,31,68,69,106]. A major drawback is their failure to capture the inherent complexity of organ systems or cancer structures. Some papers concern studies of one of the IGF1 isoforms and its derivatives (e.g., IGF1Ec, Ec peptide) in vivo and in vitro, or studies conducted only considering selected cancers (e.g., prostate cancer, thyroid cancer, NENs) [25,32,108,166]. Quantitative results of IGF1 mRNA/protein expression are not always comparable (lack of proper controls, no determination of relative expression, etc.).

The most study results point towards quantitative domination of the IGF1Ea isoform, or its comparable co-expression with the IGF1Eb isoform in cultured cancer cells: BC, CRC, EC, HCC, lung adenocarcinoma, prostate cancer [106], CC [31,106], myelogenous leukemia cells [31], human melanoma [106], as well as CRC [104,105] and CC in vivo [49]. Independent factors influence stability or growth of expression of the IGF1Ea isoform in cancer cells. In the case of CC and HCC, a role of oncogenic viruses (HPV, HCV) is suggested [31,112], while in other cancers there is a possibility of hormonal influence (testosterone), e.g., IGF1 splicing process in osteosarcoma [14]. As a reminder, in physiology only the IGF1Ea isoform is conserved among all vertebrates, whereas IGF1Eb and IGF1Ec are an evolutionary novelty originated from the exonisation of a mammalian interspersed MIR-b element [103]. The N-glycosylation site in the Ea-domain prevents its degradation in proteasomes [88].

IGF1Eb is a unique isoform, which is less conserved among primate species, and the equivalent of human IGF1Eb domain is absent in rodents [54,103]. The majority of IGF1Eb isoform expression and a switch between isoforms IGF1Eb (higher copy number) and IGF1Ea (lower copy number) was reported in HCC cells [31], and human osteosarcoma cells [31,106]. However, there are some cancer cells that do not exhibit expression of IGF1Eb, e.g., BC [106], osteosarcoma [14], or show expression only after stimulation by, e.g., DHT [14]. In tissues with CRC precancerous lesions, higher percentage of the IGF1Eb isoform was observed vs. IGF1Ea and IGF1Ec isoforms [176]. In turn, in CC, the percentage of IGF1Eb was higher than in other stages of this cancer’s development [49]. A significantly higher expression of the mRNA IGF1Eb isoform was also observed in precancerous lesions of the colon than in cancer [176].

The mechanisms of IGF1Eb isoform action in carcinogenesis are a matter of discussion and continuous research. In several types of cancer cell lines (BC, CRC, HCC) anticancer activities of the rtEa4-peptide (hEb peptide equivalent in human) were demonstrated. These properties included: dose-dependent inhibition of colony formation, promotion of cell morphology changes and cell attachment, restoration of anchorage-dependent cell division behavior, reduction of cell invasiveness, attenuation of expression of apoptotic genes in favor of cell death, and inhibition of cancer-induced angiogenesis [167,168,169,171,172,173]. In turn, other authors demonstrated growth promoting action of malignant human bronchial epithelial cells [84], axon outgrowth in neuroblastoma cells via phospho-ERK activation (MAPK pathway) [184], and enhanced cell growth of CC and osteosarcoma cells [90] after treatment with an analog of IGF1Eb precursor (Y-23-R-NH_2_) [84], recombinant rtEa4-peptide [184], or synthetic hEb [90].

There are further studies needed to fully elucidate the mechanisms and stages of Eb-peptide secretion, internalization, intracellular fate, Eb-interacting membrane molecule binding or transport to cell nucleus via IGF1R-mediated endocytosis, as well as correlation of these processes with carcinogenesis [167].

Pro-proliferative and anti-apoptotic properties, migration and invasion promotion, as well as EMT, angiogenesis, and metastasis induction mostly concern the IGF1Ec (MGF) isoform, both as the human Ec peptide [14,18,108,166,170] and IGF1Ec pro-forms [16]. These effects have been proved in hormone-sensitive breast cancer [16,170], prostate cancer [18,30,108,166], and osteosarcoma cells [14,15]. In prostate cancer cells, hEc exerts progression but not competence growth factor action, activating ERK1/2 without Akt phosphorylation [18,108]. Synthetic MGF (Ec peptide) exhibited mitogenic activity through mechanisms independent of IGF1R, INSR, and hybrid IGF1R/INSR in PC [108] and KLE cells [107]. Furthermore, in BC cells, hEc also enhances the intracellular ERK1/2 pathway [170], while osteosarcoma cell growth was stimulated via hEc peptide in an IGF1R/INSR-independent manner [14]. Moreover, an increase in cell cycle proteins (cyclin D) expression, enhanced cell migration, as well as pro-angiogenic and anti-apoptotic effects were noted, as compared to control group [15].

Difficulties in designing potential isoform IGF1 agents arise due to the complexity of IGF1 gene splicing and the presence of different factors associated with post-translational modifications, as well as bioactivity of IGF1 and its isoforms in physiology [21,23] and pathology (including cancer) [49,112,165,179]. Transcript levels not always correspond to the expression of pre-peptides, sparking suggestions to, in addition to transcripts, determine the role of IGF1 isoforms at the protein levels [31].

When it comes to regulation of various IGF1 mRNA isoform in carcinogenesis, the role of factors directly linked to IGF1 splicing, i.e., transcription factors (e.g., SP1), alternative splicing factors (e.g., FOX2) needs to be considered [49], as well as the direct role of nucleolar/nuclear localization signals of selected spliced variants of IGF1 (e.g., IGF1Eb), resulting in nuclear localization of the isoform and activation of e.g., protooncogenes [81,90,91]. Other factors include proinflammatory cytokine production in tumor microenvironment (e.g., IL-6) [17], tumor-associated viruses (e.g., HPV, HCV) [31,49,112,165,179], prolonged hypoxia and stress hormones [71], and/or other hormones (testosterone, E2, dexa) [14,30,106,108].

## 7. Concluding Remarks and Future Perspectives

Biochemical basics and biological relevance of different IGF1 mRNA isoform and post-translational processing products (E peptides) formation are only partially discovered, both in the context of physiology and pathophysiology [88,103,128].

In carcinogenesis, an important role is played mostly by paracrine/autocrine produced and released IGF1 mRNA isoforms and their derivatives. There is a hypothesis, that the action of exon 1 (class I transcripts) is preferred in this type of IGF1 activity, while exon 2 (class II transcripts) represents secretive and endocrine functions [21,68,69]. The dominance of class I transcript expression in most of the studied cancers, precancerous lesions and cell lines is confirmed by literature data [49,106,112].

While the reason for which such a variety of the IGF1 gene transcripts is “employed” for the formation of the final product, namely mature IGF1, remains unknown, available knowledge allows one to say that both transcripts and numerous IGF1 precursor forms (different than IGF1) might exhibit some activity in carcinogenesis, sometimes with antithetical effects. Further explanation is needed to elucidate the mechanisms of release of endogenous products of IGF AS, as well as their stability and other chemical properties. It is important to describe the cancer-specific molecular mechanisms of action, especially those associated with the IGF1Eb isoform (mRNA and peptide) and receptors (different than IGF1R) present on cell membranes of most common human tumors.

In the nearest perspective, we should better identify candidate therapeutic intervention targets for several IGF1 isoforms, including Eb peptide analogues exhibiting anticancer activities in certain cancers (e.g., BC, CRC, HCC) [168,169,172], as well as their antagonists, inhibiting mitogenic properties of hEB in other cancers (e.g., CC, osteosarcoma, neuroblastoma) [90,184]. An example of a compound of high therapeutic potential could be an inhibitor of IGF1Ec, the isoform of the most spectacular pro-proliferative potential in several human cancers (e.g., breast, prostate cancer, osteosarcoma). However, it needs to be noted that classical anti-GH-IGF1R would be inefficient for complete inhibition of the IGF1Ec isoform [14].

Recently, there are new hopes associated with anti-cancer therapies that use RNA binding small molecules after splicing, genetic and chemical modification of the spliceosome. Splice factor specific ASOs (group of splicing factor inhibitors) are the first FDA approved therapy to redirect splicing. Future studies in this direction will most probably concern epigenetics, accessory proteins, and RNA structures [48,190].

## Figures and Tables

**Figure 1 ijms-21-06995-f001:**
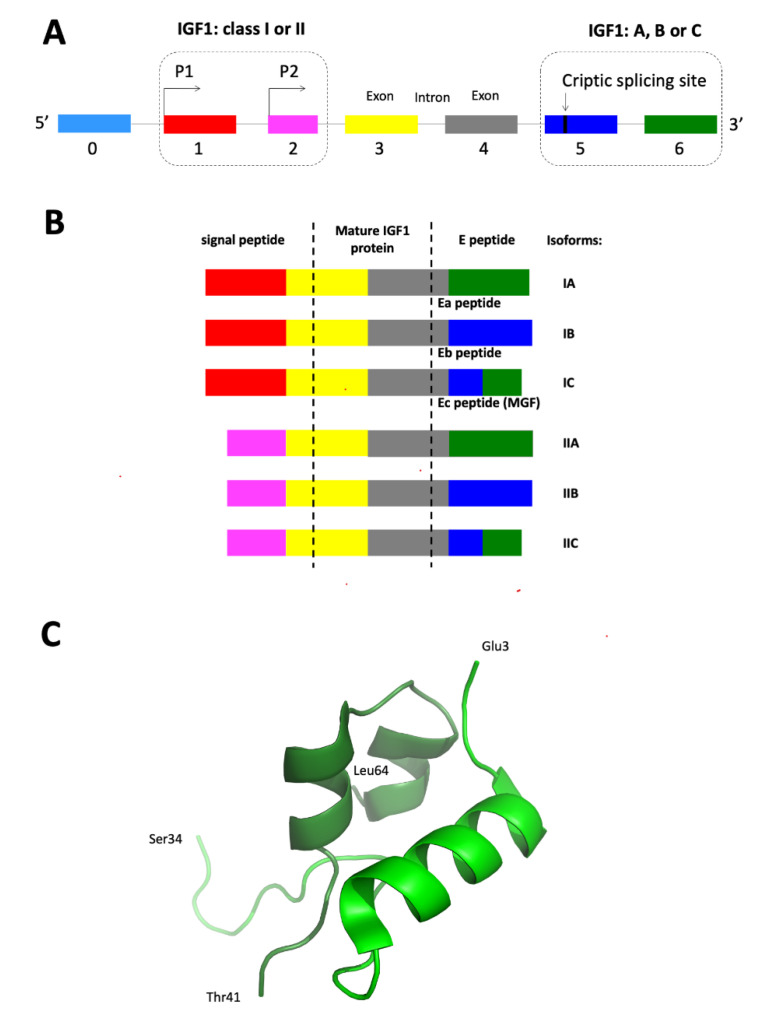
Simplified model of the human IGF1 gene structure (**A**), featuring main mRNA isoforms (variants) generated by alternative splicing and encoded precursor peptides (**B**), with the three-dimensional structure of IGF1 protein determined by X-ray crystallography, based on PDB no. 1IMX [79] (**C**). The human IGF1 gene is composed of 6 major exons and a newly discovered exon 0, upstream of exon 1 [54]. Splicing and exons in the human IGF1 gene generate distinct transcripts that vary in the 5′ and 3′ ends though the mature IGF1 protein is invariant. Transcription starts from one of the two promoters (P1 and P2) located in exon 1 and 2, respectively. Exons 1 and 2 are alternatively utilized and comprise IGF1 class I and II, respectively. Exons 3 and 4 are expressed in all known isoforms. Exon 5 is absent in isoform A (class I/II), but it forms isoforms B and C (class I/II). Transcripts containing exon 4 spliced directly to exon 6 are also referred to as IGF1Ea, those containing exon 5 spliced to exon 4 (and lacking exon 6) are referred to as IGF1Eb (unique to humans). The IGF1Ec splice variant in humans is an exon 4–5–6 variant. All peptide products derived from pro-IGF1 are shown. For details see text.

**Table 1 ijms-21-06995-t001:** In vivo expression of different IGF1 isoforms (mRNA, protein) in selected human cancers.

Human Cancer	Tissue Material/Technique	IGF1 mRNAs			IGF1 Propeptides/Peptides			No of ref.
		IGF1Ea (Class I and II)	IGF1Eb (Class I and II)	IGF1Ec (Class I and II)	Pro-Ea/Ea	Pro-Eb/Eb	Pro-Ec/Ec (MGF)	
Brain cancer	n = 9 gliomas; IHC, RT-PCR	(+)	(+)	(-)	nt			[183]
Colorectal cancer	One colonic and one retro-sigmoidal NENs; IHC	nt			nt		(-) Ec peptide in both CRC	[32]
	n = 13 pairs of CRC/CT; IHC; qRT-PCR	(+) ↑ vs. CT; ↑ vs. Eb&Ec;	(+) ↓ vs. CT; ↑ vs. Ec	(+) ≈ vs. CT	nt			[104]
		↓ class I/II vs. CT; ↑ class II vs. class I;						
	n = 28 pairs of CRC/CT; IHC; qRT-PCR	(+); 82% of all transcripts; ≈ vs. Eb; ↑ vs. Ec	(+); 17% of all transcripts; ↑ vs. Ec	(+); ~1% of all transcripts	nt			[105]
		↓ all isoforms vs. CT; CRC: Class I - 59%; class II - 41%, in quantitative expression - *NS*						
	CRC/polyps/CT; fluorescent gold nanoparticles	nt			nt		(+) Ec peptide (MGF); in C	[34]
Epithelial Cervical cancer	One uterine cervical NEN; IHC	nt			nt		(+) Ec peptide; in C	[32]
	n = 29 squamous CC (HPV+), n = 28 L-SIL (HPV+), n = 30 H-SIL (HPV+), n = 20 CT (HPV-); PCR, qRT-PCR	(+++) 85% in CC; 92% in CT	(+++) 14% in CC; 8% in CT	(+++) 1% in CC and CT	nt			[49]
		Class I - 69–86%; class II - 14–31%; ↑ of all isoforms in pre-cancerous tissues vs. CC and CT, and a shift in the balance towards IGF1Eb in CC; (+) correlation between the *FOX2* mRNA expression and Ea/Eb in precancerous and CC, and Ec in L-SIL and H-SIL						
Gastric cancer	n = 8 NENs; IHC	nt			nt		(+) Ec peptide in 37.5%; in C	[32]
Lung cancer	n = 2 NENs; IHC	nt			nt		(+) Ec peptide in one LC; in C	[32]
Pancreatic cancer	n = 17 NENs; IHC	nt			nt		(+) Ec peptide in 58.8%; in C	[32]
Prostate cancer	IHC, Western blot, RT-PCR	nt			nt		(+++) MGF; ↑ PC and PIN vs. CT	[108]
	n = 83 patients; paraffin sections; IHC	nt			nt		(+) Ec peptide; in C; #, ↑ Ec peptide	[109]
	n = 78 patients; IHC; Western blot, qRT-PCR	nt			nt		(+) Ec peptide; #, ↑ Ec peptide	[166]
Small intestine cancer	n = 9 NENs, IHC	nt			nt		(+) Ec in 44.4%; in C	[32]
Thyroid cancer	n = 92 of different types of TC; IHC, qRT-PCR	nt		(+) IGF1Ec; ↑in more aggressive vs. non-aggressive papillary TC	nt		(+) Ec peptide in papillary TC; in C; #, ↑ Ec	[35]
Urinary bladder cancer	n = 46 biopsies/CT; qRT-PCR	(+) marginally ↑ vs. CT	(+) marginally ↑ vs. CT	↓ vs. CT; #, ↓IGF1Ec	nt		nt	[33]
		↑ of all *IGF1* mRNA isoforms in in situ carcinomas						
UPO	n = 4 NENs; IHC	nt			nt		(+) Ec peptide in 100%; in C	[32]
Others	NENs; appendiceal (n = 3), gallbladder (n = 1); IHC	nt			nt		(-) Ec peptide in 100%	[32]

Legend: (+)—positive expression; (+++)—overexpression; (-)—lack of expression; ↑/↓—significant increased/decreased; #—significant correlation with clinical TNM stage, tumor grade, and/or disease recurrence; C—cytoplasm; CC—cervical cancer; CRC—colorectal cancer; CT—control tissue; HPV—Human Papillomavirus; H-SIL—high-grade squamous intraepithelial lesions; IHC—immunohistochemistry; L-SIL—low-grade squamous intraepithelial lesions; MGF—mechano-growth factor; NENs—neuroendocrine neoplasms; NS—non significant; nt—non tested; PC—prostate cancer; PIN—prostatic intraepithelial neoplasia; qRT-PCR—quantitative real-time polymerase chain reaction (PCR); TC—thyroid cancer; UPO—cancers of unknown primary origin.

**Table 2 ijms-21-06995-t002:** Differential IGF1 isoforms (mRNA, protein) expression in human cancer cell lines and mechanisms of action noted in in vitro conditions.

		IGF1 mRNA Isoforms			IGF1 Propetides/Peptides			
Human Cancer	Human Cell Lines	IGF1Ea (Class I/II)	IGF1Eb (Class I/II)	IGF1Ec (Class I/II)	Pro-Ea/Ea	Pro-Eb/Eb	Pro-Ec/Ec (MGF)	No of ref.
Breast cancer	MSF7	(+)↑ vs. Ec; class nt	(-)	(+); class nt	nt		(+) Ec peptide	[106]
		nt			nt		shEc - ↑cell proliferation and migration via ERK1/2	[170]
		nt			nt	rEb peptide - anticancer activity	nt	[168,169]
		nt			all pro-forms - ↑cell proliferation via the IGF1R; less capable of phosphorylating the IGF1R vs. mature IGF1			[16]
	MDA-MB-231	(+); class nt	(-)	(+)↑ vs. Ea; class nt	nt		(+) Ec peptide	[106]
		nt			nt		shEc did not ↑cell proliferation	[170]
		nt			nt	rEb peptide - anticancer activity	nt	[168,169,171,172]
	T47D	nt			all pro-forms - ↑cell proliferation via the IGF1R signaling, less capable of phosphorylating the IGF1R vs. mature IGF1			[16]
	ZR751	nt						
Colorectal cancer	DLD1	(+) ≈ vs. Eb, ↑ vs. Ec; from class I/II	(+)↑ vs. Ec; from class I/II	(+) from class I/II	nt		(+) Ec peptide	[106]
	SW620	nt			nt		(+) Ec (MGF) peptide	[34]
	HT29	nt			nt		(+) Ec (MGF) peptide	[34]
		nt			nt	rEb peptide - anticancer activity	nt	[169]
Endometrial cancer	KLE	(+) all mRNA isoforms in stromal cells of eutopic and ectopic endometrium; (+) IGF1Ec - in glandular cells of ectopic endometrium			nt		sEc peptide (MGF) - ↑cell growth via an IGF1R-, INSR-independent mechanism	[107]
		(+)↑ vs. Eb; from class I/II	(+); from class I/II	(+)↑ vs. Ea&Eb; from class I/II	(+) pro-Ea; ↑ vs. other cells	nt	(+) Ec peptide; ↑ vs. other cells	[106]
Epithelial Cervical cancer	HeLa (HPV18+)	nt			nt	shEb peptide - ↑cell growth; in N	nt	[90]
		(+)↑ vs. Eb&Ec; in C	(+)↑ vs. Ec	(+)	(+) pro-IGF1A; ↑ vs. other cells	(+) pro-IGF1B in N; (+) hEb peptide in N	(+) Ec - very low expression	[31]
		(+) ≈ vs. Eb; from class II	(+) from class I	(+)↑ vs. Ea&Eb; from class I	nt		(+) Ec peptide; ↑ vs. other cells	[106]
Hepatocellular cancer	HepG2	(++)↑ vs. Ec	(+++)↑ vs. Ea and Ec	(+)	(+) pro-IGF1A; (-) Ea peptide	(-) pro-IGF1B;(+) hEb peptide in N	(+) Ec - very low expression	[31]
		nt			nt	rEb - anticancer activity	nt	[169]
	HuH7	(+) ≈ vs. Eb; from class I/II	(+) from class I	(+)↑ vs. Ea&Eb; from class I	nt		(+) Ec peptide	[106]
Lung cancer	NCI-H345	nt			nt	sEb peptide - ↑cell proliferation via an IGF1R-independent mechanism	nt	[84]
	A549	(+)↑ vs. Eb; from class I/II	(+) from class I	(+)↑ vs. Ea&Eb; from class I/II	nt		(+) Ec peptide	[106]
Melanoma malignum	SK-MEL28	(+)↑ vs. Eb&Ec; from class I/II	(+)↑ vs. Ec; ↑ vs. other cells; from class I	(+); from class I	nt	(+) pro-IGF1Eb; ↑ vs. other cells	(+) Ec peptide; ↑ vs. other cells	[106]
Osteosarcoma	U2OS	(+)↑ vs. Ec	(+)↑ vs. Ea and Ec	(+)	(+) pro-IGF1A; (-) Ea peptide	(-) pro-IGF1B;(+) hEb in N	(+) Ec - very low expression	[31]
		nt			nt	sEb peptide - ↑cell growth; in N	nt	[90]
	MG63	(+) from class I	(+)↑ vs. Ea; from class II	(+)↑ vs. Ea&Eb; from class I	nt		(+) Ec peptide; ↑ vs. other cells	[106]
		nt		(+) Ec - ↑ vs. MHos cells	nt		sEc peptide (MGF) - ↑cell proliferation and migration	[15]
		(+)	(-); (+) after exposure to DHT for 72 h	(+)	nt		sEc peptide - ↑cell growth via IGF1R/INSR/hybrid receptor-independent way	[14]
Prostate cancer	LnCaP	(+);↑ vs. Eb; ↑ vs. other cells; from class I/II	(+)↑ vs. other cells; from class I	(+)↑ vs. Ea&Eb; from class I	nt		(+) Ec peptide; ↑ vs. other cells	[106]
		nt		(+)	nt		(+) Ec peptide; sEc peptide - ↑cell growth via ERK1/2 and IGF1R/INSR/hybrid receptor-independent mechanism	[108]
	PC3	nt		(+)	nt		(+) Ec peptide; sEc - ↑cell growth via ERK1/2 and IGF1R/INSR/hybrid receptor-independent mechanism	[108]
		nt		(+)	nt		(+) Ec peptide; endogenous Ec peptide - ↑cell proliferation via ERK1/2	[166]
Myelogenous leukemia	K562	(+++)↑ vs. Eb&Ec	(++)↑ vs. Ec	(+)	(+) pro-IGF1A; (-) Ea peptide	(-) pro-IGF1B; ↑ Eb peptide vs. other cells; (+) hEb in N	(+) Ec - very low expression	[31]

Legend: (+)—positive expression; (++)—high expression; (+++)—overexpression; (-) lack of expression; ≈—comparable; ↑—higher or increased; &—and; C—cytoplasmic fraction/localization; DHT—dihydrotestosterone; h—hours; HPV—Human Papillomavirus; N—nucleus/nucleolus; nt—non tested; rEb—recombinant IGF1Eb isoform; shEb—synthetic human Eb peptide; sMGF—synthetic mechano-growth factor

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
