# Peer review of "Role of Alternatively Spliced Messenger RNA (mRNA) Isoforms of the Insulin-Like Growth Factor 1 (IGF1) in Selected Human Tumors"

_ijms, 2020, doi:10.3390/ijms21196995_

Round 1

Reviewer 1 Report

This manuscript appears as well-written overview on the current state of knowledge in the field. It raises interesting research questions and addresses future research topics.

Author Response

Dear Reviewer,

We wish to thank you very much for a favourable review, positive evaluation of the work, and time spent on reviewing the manuscript.

All minor changes and/or additions that have been noticed in the text are made and marked in red color.

As per suggestion, the manuscript was thoroughly revised, with all language errors corrected. The publication was corrected by a qualified, native speaker, familiar with the manuscript topics.

Sincerely yours,

Aldona Kasprzak

Reviewer 2 Report

This review article summarized the most recent updates regarding the alternative splicing of mRNA isoform of IGF1. The presence of different IGF1 transcripts suggests tissue-specific action and the mechanistic explanation of the possible biological role of different variants of IGF1 mRNAs and pro-peptides on carcinogenesis is well described.

Author Response

Dear Reviewer,

We wish to thank you very much for a favourable review, positive evaluation of the work, and time spent on reviewing the manuscript.

All minor changes (and all additions) that have been noticed in the text are made and marked in red color.

Additionally the publication was corrected by a qualified, native speaker, familiar with the manuscript topics.

Sincerely yours,

Aldona Kasprzak
